# Olive Oil-Based Oleogel as Fat Replacer in a Sponge Cake: A Comparative Study and Optimization

**DOI:** 10.3390/foods11172643

**Published:** 2022-08-31

**Authors:** Francesca Malvano, Mariachiara Laudisio, Donatella Albanese, Matteo d’Amore, Francesco Marra

**Affiliations:** 1Department of Industrial Engineering, University of Salerno, 84084 Fisciano, SA, Italy; 2Department of Pharmacy, University of Salerno, 84084 Fisciano, SA, Italy

**Keywords:** oleogel, mixture design, DOE, bakery products

## Abstract

Oleogels (defined as structured solid-like materials with a high amount of oil entrapped within a three-dimensional network of gelator molecules) represent a healthy alternative to fats that are rich in saturated and trans fatty acids. Given its fatty acids composition (oleic, linoleic, and linolenic acids), olive oil is an excellent candidate for the use of oleogels in the food industry. In this study, a D-optimal mixture design was employed to optimize the replacement of butter with olive oil-based oleogel in a type of sponge cake formulation: the plum cake. In addition, emulsifiers and whey proteins were used as recipe ingredients to extend the product’s shelf life by delaying staling phenomena and mold growth. In the experimental design, oleogel, emulsifier, and whey protein variables were set as the ingredients that change in specific ranges, while hardness, porosity, water activity, and moistness were used to characterize the obtained formulations. The experimental data of each response were fitted through polynomial regression models with the aim of identifying the best plum cake formulation. The results revealed that the best mixture was the formulation containing 76.98% olive oil-based oleogel, 7.28% emulsifier E471, and 15.73% whey protein. We stored the optimized plum cake for 3 months at room temperature and then checked for any hardness and moistness changes or mold spoilage.

## 1. Introduction

Sponge cakes are air-leavened bakery products that give rise to a wide range of traditional sweet bakery products consumed daily by people in the world. They are characterized by a high amount of saturated fats, ranging from 7.5% to 31%, that greatly influence the nutritional and functional characteristics of the final product [1].

The incorporation of air and the stabilization of air bubbles by the fats result in an increase in the volume of the loaf and a uniform cellular structure. Moreover, the lubricating property of fats guarantees softness to the final product, in addition to preventing drying and retaining moistness during storage. Other important functions of fat in sponge cake products are the greasiness of the finished product and the oxidative stability during storage [2].

However, the negative effects of saturated and trans-fat consumption on human health have been widely investigated. As part of the effort to reduce saturated fatty acids, oleogels appear to provide a healthy and promising alternative. Oleogels can be defined as structured solid-like materials with a high amount of oil entrapped within the three-dimensional network of gelator molecules. Since the liquid oil contains a relatively low amount of saturated fat, replacing solid fat with oleogel in processed foods would certainly lead to an improvement in the nutritional properties [3].

Recently, the use of this approach has gained considerable interest in the bakery industry [4]. As a shortening alternative in cookie formulation, different wax-based oleogels with vegetable oils were investigated [5,6,7,8]. Olive oil is one of the most promising vegetable oil for oleogel production due to the positive health effects from its fatty acids compositions (oleic, linoleic, and linolenic acids), in addition to minor components (tocopherols and phytosterols) [9].

Natural wax oleogels with different vegetable oils were also used as fat replacers in baked cake formulation: Oh et al. [10] showed that beeswax-based oleogels were the most effective in producing cakes with higher nutritional properties, compared with cakes prepared with shortenings without changes in quality attributes. Moreover, the partial shortening replacement with carnauba wax oleogels at up to 50% was effective in preserving the ability to retain air cells in cake dough [11].

Recently, Giacomozzi et al. [12] and Oh and Lee [13] highlighted the effectiveness of partially replacing margarine with monoglycerides and hydroxypropyl methylcellulose-based oleogels for muffin production.

Despite their widespread consumption, baked goods are characterized by a limited shelf-life at room temperature, undergoing physical and chemical changes during storage, in addition to mold spoilage [14]. Physical and chemical changes, resulting in the loss of freshness and the impairment of texture, are related to the staling phenomenon, which implies a leathering of the crust and a hardening of the crumb. The major cause of crust staling is moisture adsorption from the air and the redistribution of moisture from the interior crumb to the crust [15]. Another cause of staling is the retrogradation of starch, a process that occurs in gelatinized starch as it moves from an initial amorphous state to a more ordered or crystalline state, resulting in an unacceptable increase in the firmness of foodstuffs [16]. Mono- and diglycerides of fatty acids (E471) may be used to overcome bread staling since they create emulsified complexes with amylose, ensuring a soft crumb structure during the storage, in addition to acting as preservatives [17]. Whey proteins, apart from being highly nutritious, are useful food ingredients in cakes [18,19]. In particular, whey proteins form gels capable of holding water, reducing free water available for mold growth [19], which is one of the major concerns regarding the shelf life of bakery products.

Mixture design is a common statistical method used to investigate the function of ingredients in processed food with the aim of optimizing their formulation [20]. Mixture design methods based on mathematical models, aimed to link consumers’ preferences with product formulation, can contribute to the development of digital tools for food product design [21].

The scientific literature reports several mixture design applications for the optimization of bakery product formulations. Some of these include the optimization of gluten- free flours for gluten-free bread formulation [22], tikhur starch as semolina substitute in the preparation of baked milk cake [23], and yacon and maca flour in the formulation of chocolate cake [24].

The aim of this study was to identify the best formulation of a mixture prepared with olive oil-based oleogel, an emulsifier and whey proteins, as a replacement for butter in a popular European sponge cake formulation: the plum cake. A D-optimal mixture design was used to determine the effect of different formulations on the main physicochemical and sensory properties affecting the quality of plum cakes. Finally, the optimized plum cake was packed and stored to evaluate the shelf-life by means of staling and mold-spoilage measurements.

## 2. Materials and Methods

### 2.1. Materials

The basic ingredients of the plum cake’s dough, such as wheat flour type 00, sugar (sucrose), whole natural white yogurt, eggs, olive oil, and chemical leavening (disodium diphosphate and sodium hydrogen carbonate) were bought in a local market. Beeswax (BW) was purchased at ACEF (Piacenza, Italy), while emulsifier E471 (mono- and diglycerides of fatty acids) was purchased at SaporePuro (Torino, Italy).

### 2.2. Experimental Design

The control plum cake recipe was obtained using preliminary experiments focused on the selection of the ratio among the ingredients listed in cake recipe books. We performed a sensory evaluation of the produced plum cakes by structure and taste.

The selected plum cake formulation was modified by replacing butter with different amounts of olive oil-based oleogel and by adding emulsifier E471 and whey proteins. A D-optimal mixture design was used to optimize the experiments, which were carried out to evaluate the effect of olive oil-based oleogel (*x*_1_), emulsifier E471 (*x*_2_), and whey proteins (*x*_3_) on the physicochemical and sensory properties of plum cakes.

In the mixture experiment design, the ingredients’ composition was expressed as the relative mass fraction, defined as in Equation (1):(1)$$x_{i}=m_{i}/\left(m_{1}+m_{2}+m_{3}\right)$$
where *m*_1_, *m*_2_, and *m*_3_ represent the masses of the olive oil-based oleogel, the emulsifier, and the whey proteins, respectively. The sum of the relative mass fractions being 1,
(2)$$\sum_{i=1}^{3}\;x_{i}=x_{1}+x_{2}+x_{3}=1$$

The range of mass fraction of the single ingredients in the mixture was established as follows, based on preliminary experimental tests: 0.65%–1 for olive oil-based oleogel, 0–0.1 for emulsifier E471, and 0–0.25 for whey proteins. Table 1 presents the experimental design comprising 10 different formulations. The responses, or dependent variables, that were evaluated for each formulation were: three physical properties (hardness, *y*_1_; water activity, *y*_2_; porosity, *y*_3_) and one sensory property (moistness, *y*_4_).

A graphical display of the mixture design is shown in the ternary plot in Figure 1, where each corner of the triangle corresponds to an independent variable, the white area represents the experimental region, and the black points inside are the formulations.

### 2.3. Oleogel Preparation

The beeswax (BW) (3%) was dissolved in olive oil at 85 °C under magnetic stirring at 100 rpm. After the complete dissolution, the mixture was cooled to room temperature (25 ± 2 °C) for the formation of oleogels. The oleogel was stored at room temperature for at least 24 h before being utilized.

### 2.4. Sample Preparation and Characterization

The plum cake formulation included 29% 00 flour, 25% sugar, 8% yogurt, 20% butter, 18% eggs, and chemical leavening. The plum cake dough was produced by first whipping eggs and sugar; next, butter was added and mixed. Finally, yogurt, flour, and chemical leavening were mixed until we obtained a smooth and homogeneous dough. The dough was transferred into paper cake molds (3.3 cm × 8.5 cm × 2.6 cm) and baked in a hot air oven for 8 min at 160 °C and then for 5 min at 150 °C.

#### 2.4.1. Crumb Porosity and Water Activity Evaluation

The evaluation of the morphological features of plum cakes was performed using an image analysis system. The captured images were analyzed using Image-Pro Plus 5.1 software (Media Cybernetic, Rockville, USA), and the porosity parameter was evaluated. Crumb porosity was expressed as the mean value of the total cell to the total area ratio on the different slices analyzed.

The water activity (a_w_) of the plum cakes was determined using a water activity meter (Testo 650, Testo Inc., West Chester, PA, USA) at 25 °C.

#### 2.4.2. Texture Profile Analysis

The texture profile analysis (TPA) of plum cakes was carried out using a texturometer (LRX Plus, Lloyd Instruments, Chicago) equipped with a load cell of 50 N. The samples were submitted to a double compression of 50% with a flat-end cylindrical probe at a speed of 1.0 mm/s. The parameters investigated through the TPA analysis were hardness, cohesiveness, springiness, and chewiness. The data are reported as the average of 5 measurements carried out on 5 plum cakes.

#### 2.4.3. Sensory Analysis

The sensory properties of the plum cake samples were measured according to Albanese et al. [25]. Ten panelists were chosen from among the students and professors of the Department of Industrial Engineering at the University of Salerno (Italy) based on their ability to identify odors and the five basic tastes. The parameters taken into consideration, rated on 5-point scales from “none” (1) to high” (5), were moistness (the wet sensation of cake products during swallowing), sweetness, flavor, off flavor, and overall acceptability.

### 2.5. Statistical Analysis

JMP statistical software (SAS Institute. Inc. Cary, NC, USA) was used to constitute the mixture design, to analyze the experimental data, and to optimize the plum cake’s formulation.

The Scheffè canonic equation was used to model the experiment output in terms of physical properties (hardness, porosity, and water activity) and the sensory attribute (moistness), particularly, for the generic output:(3)$$y_{i}=\beta_{i1}x_{1}+\beta_{i2}x_{2}+\beta_{i3}x_{3}+\beta_{i12}x_{1}x_{2}+\beta_{i13}x_{1}x_{3}+\beta_{i23}x_{2}x_{3}+\beta_{i123}x_{1}x_{2}x_{3}$$
where *y_i_* is the studied response; $\beta_{i1}$, $\beta_{i2}$, $\beta_{i3}$, $\beta_{i12}$, $\beta_{i13}$, $\beta_{i23}$, and $\beta_{i123}$ are the regression parameters; and *x*_1_, *x*_2_, and *x*_3_ are the independent variables.

The statistical parameters used in evaluating and selecting the best-fitted model were the coefficient of determination (*R*^2^), adjusted coefficient of determination (adjusted *R*^2^), standard deviation, and lack-of-fit and regression data (*p* value and *F* value, respectively). From the model equation, the positivity of the coefficient represented the positive contribution toward the response and vice versa. Triangular contour plots, generated from the polynomial equation for each response variable, were superimposed by means of the desirability function. A desirability function approach is widely used in the optimization of the mean of multiple responses. In particular, the optimum formulation was obtained by setting the goal of the desirability functions for the hardness and water activity to a minimum value and for the porosity and moistness to a maximum value in established value ranges through the measurement of the physical and sensory parameters of a plum cakes market leader. The other sensory properties (sweetness, flavor, off flavor, and overall acceptability) were used for the global evaluation of the final product.

All the analyses were performed in triplicates. The experimental data were reported as the mean and standard deviation and subjected to analysis of variance (ANOVA). The significant differences (*p* < 0.05) among the plum cake samples, as well as between the plum cakes after the 3 months of storage, were determined using a Student’s *t*-test with SPSS software version 13.0 for Windows (SPSS, Inc., Chicago, IL, USA).

### 2.6. Plumcake Storage

The plum cakes, realized under optimum formulation, were single-packed in an impermeable plastic film and stored at room temperature for 3 months. At the end of storage, we evaluated their hardness, water activity, porosity, and moistness to verify the efficacy of formulation on the shelf life of the product.

## 3. Results and Discussion

Table 2 shows the experimental design with independent variables and the respective responses.

The experimental data were statistically analyzed to identify the best-fitted model for each independent variable. The statistical parameter values, obtained for each model, are shown in Table 3. All the models were significant (*p* < 0.05), and the lack of fit was insignificant. Thus, the possibility of error occurring was low.

In order to underline the effect of each ingredient and its interactions on the responses, the estimated parameters of the prediction models for each response are also reported in Table 3. When the independent variable significantly (*p* < 0.05) affects the response, the value is listed in bold. A positive term in the regression equation represents an effect that favors optimization due to synergistic effects, whereas a negative term reveals an antagonistic effect between the factors and the response [26].

All the responses were significantly (*p* < 0.05) affected by the presence of the olive oil-based oleogel, as well as by the addition of the whey proteins to the control plum cake’s formulation.

In particular, the hardness, porosity, and moistness were mainly affected by the linear terms of the oleogel and whey proteins. The water activity was influenced both by the linear term of the emulsifier and the quadratic terms related to the interaction between the oleogel and whey proteins.

For a better understanding and to graphically view the relationship between the percentage of the ingredients and each response, 2D-contour plots of the fitted polynomial regressions are shown in Figure 2.

Figure 2a shows that the lowest hardness values were reached in the formulations with the high oleogel and whey protein amounts. The presence of the whey proteins improved the aeration of the dough, increasing the porosity of the samples (Figure 2b).

Conversely, the combined presence of the emulsifier and whey proteins affected the water activity of the samples, which decreased when the amounts increased (Figure 2c). Finally, the moistness parameter reached the highest values with a high amount of olive oil-based oleogel.

### 3.1. Influence of Fat Replacement with Olive Oil-Based Oleogel on the Plum Cake’s Texture Profile

The influence of the replacement of butter with olive oil-based oleogel on the plum cake’s texture profile was evaluated (Figure 3).

A comparison among the control and oleogel-replaced formulations highlighted that the hardness values of plum cakes with olive oil-based oleogel were, for each formulation, always lower than the control sample formulated with butter (Figure 3a). Our results are in agreement with those from the studies by Aliasl Khiabani et al. [27] and Yilmaz and Ogutcu [5], who observed a reduction in the hardness values of bakery products formulated with carnauba wax and beeswax-based oleogels, respectively, in comparison with the control formulated with commercial bakery shortening.

The lower hardness values obtained in the samples formulated with a high percentage of olive oil-based oleogel (ranging from 78–90%) with respect to the other samples could be also attributed to the presence of a high percentage of emulsifier and whey proteins. Emulsifiers, in fact, are commonly used in leavened baked products as binding agents to improve gas retention capacity, improve and soften the porous crumb structure, and obtain anti-staling effects [28]. A study carried out by Onyango et al. [29] highlighted a decrease in crumb hardness, as well as the staling rate, with the increase of the emulsifier concentration. Thanks to their excellent functional properties, such as foaming capability, emulsifying properties, and gel-formation ability [30], whey proteins are largely investigated in bakery products, underlining the significant contribution in creating smaller air cells resulting in high foam stability [31].

According to the hardness results, the springiness increased with the increase of the oleogel and emulsifier amounts. The same trend was recorded for gumminess. On the contrary, the chewiness, which represents the amount of energy needed to disintegrate a food for swallowing, decreased as the oleogel percentage increased.

### 3.2. Crumb Structure Evaluation

The effect of butter replacement with the olive oil-based oleogel on the aeration of the plum cake structure was investigated in terms of porosity. Fats, in fact, play an important role in the aeration of leavened products: the air cells are incorporated into the fat phase during the mixing step and then released into an aqueous phase when the fat is melted during baking, consequently giving rise to a foam structure [9]. According to the literature [10,12], the total porosity of the plum cake samples ranged from 73.32% to 89.05%, with the highest value presented by the sample formulated with oleogel 72%, emulsifier 3%, and whey proteins 25% (Figure 4a). As reported by Rathnayake et al. [28] a produc t with a well-developed porous crumb structure should have high porosity and a fine, regular gas cell structure.

Previous studies [9,11,12,32] highlighted that replacing shortening with oleogel in leavened baked products involved a lower amount of air incorporated into the dough and, consequently, a lower total porosity, negatively contributing to the product’s final quality. However, the addition of whey proteins and emulsifier, which improved the aeration of the oleogel-replaced dough, allowed for high porosity values with a pore diameter close to those of the control sample (Figure 4b).

As expected, reduced porosity values were registered in samples formulated without whey proteins and emulsifier, likely due to the lower air-retaining capacity of the dough, causing a denser structure.

The pore size of all samples ranged from 0.19 mm to 0.29 mm, with the lowest value found in the plum cake formulated with 72% olive oil-based oleogel, 3% emulsifier, and 25% whey proteins.

The size of the air cell is an important factor that affects the texture of the final product. As reported in the literature [11], larger air cells in crumbly baked goods could determine a more crumbly product and the loss of its typical shape, recognized as negative quality features. Moreover, a crumbly structure promotes the oil’s migration toward the crust.

As reported by Calligaris et al. [33], in muffins produced with monoglyceride-based oleogels prepared with palm oil and sunflower oil, a high percentage of small air cells generated a more homogenous crumb. It seems that monoglyceride aided in the transformation of large air cells into small and uniform cells, which results in a crumb with a finer texture.

### 3.3. Water Activity Evaluation

Water activity (a_w_) is often considered the most effective factor in inhibiting mold growth [34].

With the aim of delaying mold spoilage and thus increasing the shelf life of the product, the a_w_ values of all butter-replacement formulations and of the control sample were evaluated.

The comparison of a_w_ values among the control (0.83 ± 0.01) and all butter-replacement formulations (Table 2) showed no significant (*p* < 0.05) differences except for the sample containing 72% olive oil-based oleogel, 3% emulsifier, and 25% whey proteins, which showed the lowest aw value of 0.79.

According to Goncalves et al. (2017) [35], whey proteins in leavened product formulations possess excellent water-binding properties that might be the reason for the detected low water activity.

### 3.4. Evaluation of Sensory Attributes

The sensory profile of all plum cake samples is shown in Figure 5.

The results highlighted that when butter is replaced with the mixture of oleogel/emulsifier/whey proteins, a significant (*p* < 0.05) decrease in the moistness value was reached, compared with that of the control (Figure 5). However, the higher that the oleogel content was, the greater the moistness was of the samples. The addition of the maximum amount of whey proteins (25%) and emulsifier (10%) to the butter determined the lowest moistness values and, consequently, produced a product with low overall acceptability.

Negligible differences among samples were recorded regarding sweetness, and no formulation showed off-flavors.

In conclusion, the formulation with the highest overall acceptability was oleogel 78%, emulsifier 10%, and whey proteins 12%.

### 3.5. Mixture Optimization to Produce a Plum Cake with Desired Characteristics

The mathematical models obtained for each response were used to carry out the optimization of the plum cake formulation. The goal of the optimization was to determine the best ratio of olive oil-based oleogel, E471, and whey proteins in order to obtain a product with the proper textural and sensory properties, as explained in Section 2.5.

The optimum percentages of ingredients, as generated by the JMP software through the desirability function, were 76.98% olive oil-based oleogel, 7.28% emulsifier E471, and 15.73% whey proteins.

Plum cakes with the optimized formulation were produced in order to verify a match between the predicted and observed values of hardness, water activity, porosity, and moistness. The predicted values were obtained by applying Equation (3) to each property (Appendix A).

It is worth noting that the predicted value of each response was calculated on the plum cake samples before the storage period, with (t = 0) being the model based on the physicochemical and sensory parameters that changed during the storage period.

Finally, with the aim of verifying the efficacy of formulation best able to preserve the investigated quality parameters, we identified the hardness, porosity, a_w_, and moistness before and after 3 months of storage, and these factors are reported in Table 4.

The stored plum cakes showed an a_w_ decrease, resulting in an increase in hardness and a decrease in moistness. Despite these changes, the measured water activity prevented the development of mold during the storage period, likely due to the properties of the emulsifier and whey proteins that were added to the dough.

## 4. Conclusions

This work investigated the replacement of butter with olive oil-based oleogel in plum cake formulation, proposing a mixture design approach for optimal product formulation. A D-optimal mixture design was used to identify the best plumbcake formulation in order to obtain a product with textural and sensory properties close to those of the control formulation. By the experimental design, 10 different formulations were analyzed in terms of their hardness, water activity, porosity, and moistness. The statistical analysis of the results highlighted a decrease in plum cake hardness with an increase in oleogel and whey proteins. Moreover, the presence of whey proteins improved the aeration of the batter, increasing the porosity of the samples. Conversely, the combined presence of the emulsifiers and whey proteins affected the water activity of the samples, which decreased when the emulsifiers and whey proteins increased. Finally, moistness reached its highest values in formulations with a high amount of olive oil-based oleogel.

The results of the mixture optimization revealed that the best mixture was the formulation containing 76.98% olive oil-based oleogel, 7.28% emulsifier E471, and 15.73% whey proteins, according to the desirability function approach. A three-month storage study at room temperature was carried out to identify the optimum plum cake formulation through observing changes in the hardness and moistness parameters and a lack of mold spoilage, suggesting a long shelf life.

## Figures and Tables

**Figure 1 foods-11-02643-f001:**
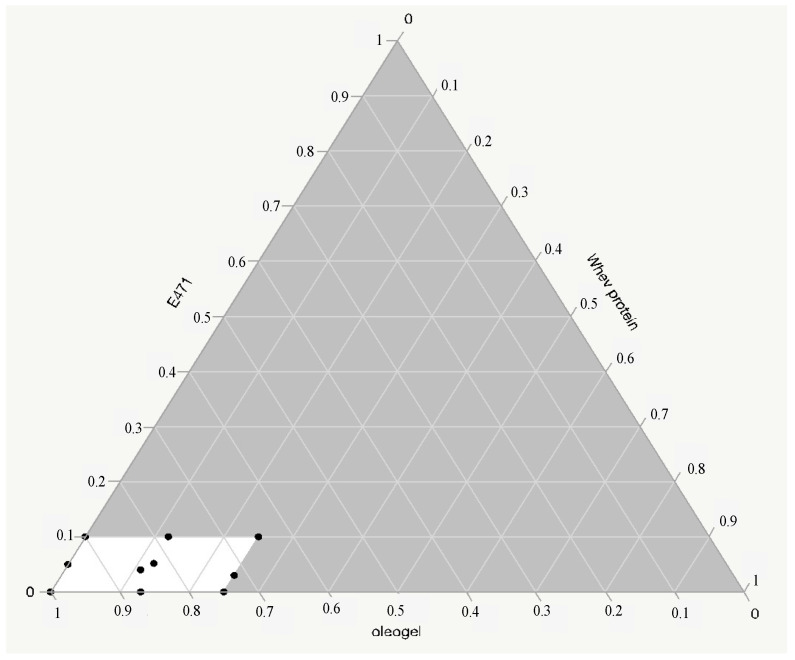
Ternary plot of mixture design.

**Figure 2 foods-11-02643-f002:**
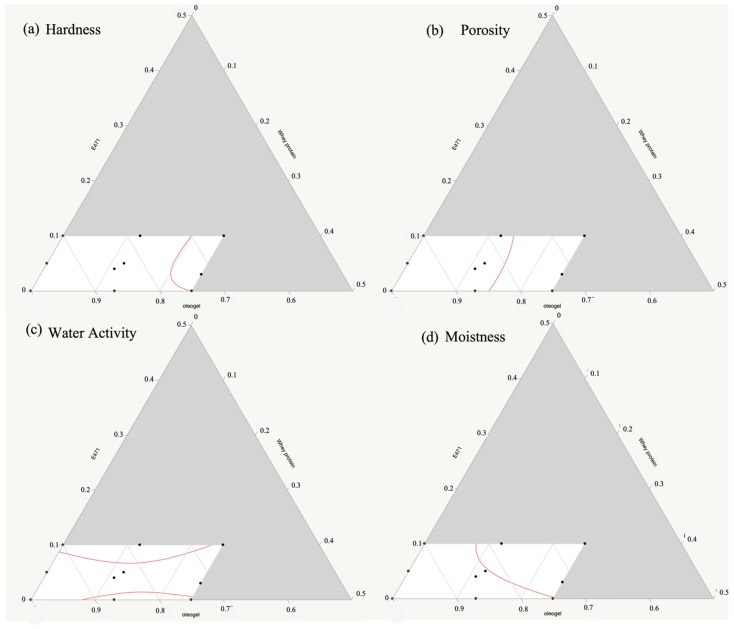
2D-contour plots: the effects of the oleogel, emulsifier E471, and whey proteins on the hardness (**a**), porosity (**b**), water activity (**c**), and moistness (**d**). Black dots correspond to the plumcake formulations, red lines correspond to the regression equations for each response.

**Figure 3 foods-11-02643-f003:**
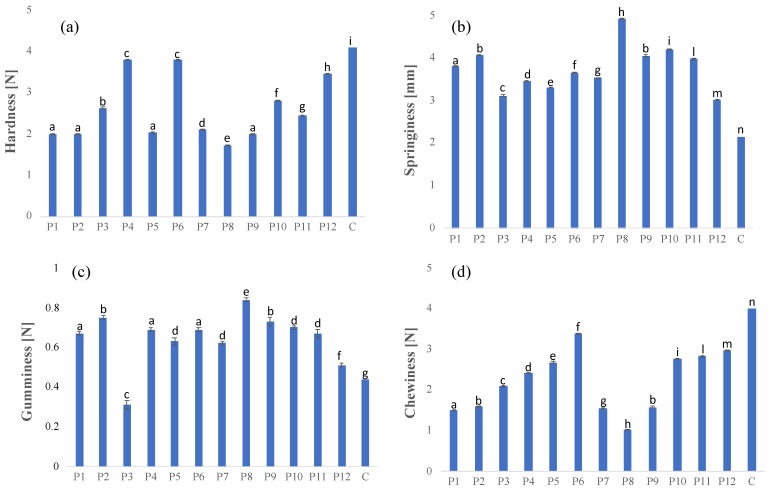
Hardness (**a**), springiness (**b**), gumminess (**c**), and chewiness (**d**) results. Different letters (a, b, etc.) reveal significant differences (*p* < 0.05) among the samples.

**Figure 4 foods-11-02643-f004:**
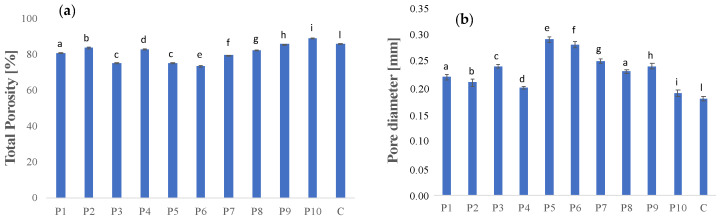
Total porosity (**a**) and pore diameter (**b**) of the butter replacement and control samples. Different letters (a, b, etc.) reveal significant differences (*p* < 0.05) among the samples.

**Figure 5 foods-11-02643-f005:**
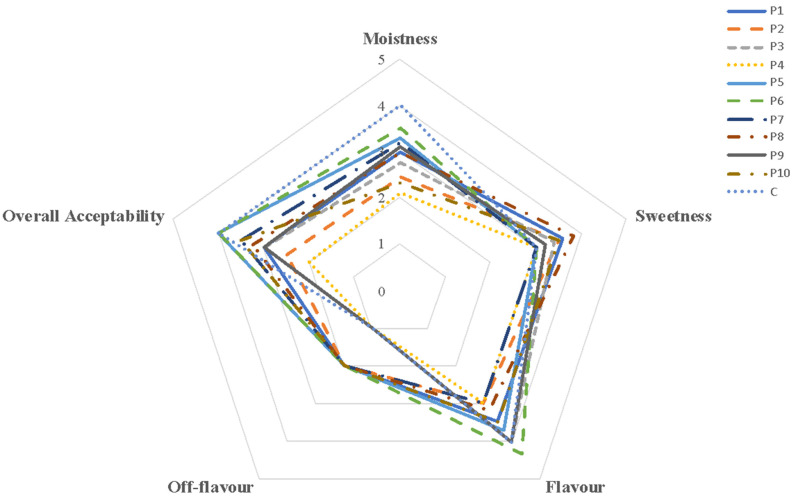
Spider plots of sensory attributes (moistness, sweetness, flavor, off-flavor, and overall acceptability) of plum cake samples.

**Table 1 foods-11-02643-t001:** D-optimal mixture design applied to optimize the plum cake formulation.

Formulation	Coded Ingredients			Original Ingredients (g/100 g_dough_)		
	*x* _1_	*x* _2_	*x* _3_	Olive Oil Oleogel	Emulsifier E471	Whey Proteins
P1	0.85	0.04	0.11	17.00	0.80	2.20
P2	0.78	0.10	0.12	15.60	2.00	2.40
P3	0.75	0.00	0.25	15.00	0.00	5.00
P4	0.65	0.10	0.25	13.00	2.00	5.00
P5	1.00	0.00	0.00	20.00	0.00	0.00
P6	0.83	0.05	0.12	16.60	1.00	2.40
P7	0.90	0.10	0.00	18.00	2.00	0.00
P8	0.87	0.00	0.13	17.40	0.00	2.60
P9	0.95	0.05	0.00	19.00	1.00	0.00
P10	0.72	0.03	0.25	14.40	0.60	5.00

**Table 2 foods-11-02643-t002:** Experimental data for each response.

Formulation	Ingredients’ Composition			Hardness [N]	Water Activity	Porosity [%]	Moistness
	*x* _1_	*x* _2_	*x* _3_	*y* _1_	*y* _2_	*y* _3_	*y* _4_
P1	0.85	0.04	0.11	2.00 ± 0.01 ^a^	0.82 ± 0.01 ^a^	80.60 ± 0.20 ^a^	3.00 ± 0.01 ^a^
P2	0.78	0.10	0.12	2.00 ± 0.01 ^a^	0.81 ± 0.01 ^a^	83.68 ± 0.40 ^b^	2.45 ± 0.02 ^b^
P3	0.75	0.00	0.25	2.64 ± 0.02 ^b^	0.81 ± 0.01 ^a^	74.87 ± 0.30 ^c^	2.78 ± 0.01 ^c^
P4	0.65	0.10	0.25	3.80 ± 0.01 ^c^	0.81 ± 0.01 ^a^	82.78 ± 0.10 ^d^	2.12 ± 0.02 ^d^
P5	1.00	0.00	0.00	2.04 ± 0.02 ^a^	0.84 ± 0.01 ^a^	75.17 ± 0.20 ^c^	3.30 ± 0.01 ^e^
P6	0.83	0.05	0.12	2.12 ± 0.01 ^d^	0.84 ± 0.01 ^a^	73.32 ± 0.20 ^e^	3.50 ± 0.03 ^f^
P7	0.90	0.10	0.00	1.73 ± 0.01 ^e^	0.84 ± 0.01 ^a^	79.46 ± 0.30 ^f^	3.20 ± 0.01 ^g^
P8	0.87	0.00	0.13	2.81 ± 0.01 ^f^	0.81 ± 0.02 ^a^	82.34 ± 0.30 ^g^	3.00 ± 0.01 ^a^
P9	0.95	0.05	0.00	2.45 ± 0.02 ^g^	0.84 ± 0.02 ^a^	85.62 ± 0.10 ^h^	3.10 ± 0.01 ^h^
P10	0.72	0.03	0.25	3.47 ± 0.01 ^h^	0.79 ± 0.01 ^b^	89.05 ± 0.30 ^i^	2.34 ± 0.01 ^i^

Different letters (^a, b, c^…) in the same column reveal significant differences (*p* < 0.05) among the samples.

**Table 3 foods-11-02643-t003:** Statistical parameters and estimated parameters for independent variables in the prediction models for each response. When the variable significantly affects the response, the values are listed in bold italics (*p* < 0.05).

	Hardness		Water Activity		Porosity		Moistness	
Model	Significant		Significant		Significant		Significant	
R^2^	0.9274		0.9581		0.7956		0.9639	
Adjusted R^2^	0.8404		0.9078		0.7623		0.9206	
*p* value	0.0100		0.0027		0.0109		0.0019	
F value	10.6547		19.0481		1.2274		22.2434	
Lack of fit	Not significant		Not significant		Not significant		Not significant	
Standard deviation	0.2970		0.0051		4.5004		0.1359	
	Estimated parameter ($\beta_{i}$)	*p*-value	Estimated parameter ($\beta_{i}$)	*p*-value	Estimated parameter ($\beta_{i}$)	*p*-value	Estimated parameter ($\beta_{i}$)	*p*-value
Oleogel	**2.1141**	0.0002	**0.8396**	<0.0001	**74.6728**	<0.0001	**3.3794**	<0.0001
E471	−11.2812	0.2124	**0.7520**	0.0026	−132.5664	0.3183	4.6596	0.2537
Whey proteins	**3.1417**	0.0106	**0.8099**	<0.0001	**75.7432**	0.0015	**2.3239**	0.0014
Oleogel*E471	17.1307	0.1762	0.1216	0.5435	316.2800	0.1133	−2.9226	0.5829
Oleogel*whey proteins	0.1614	0.9460	**−0.1062**	0.0411	14.5741	0.6889	0.4761	0.6654
E471*whey proteins	23.6151	0.1087	0.07898	0.7195	315.6269	0.1462	−4.5795	0.4464
Oleogel*E471*whey proteins	−23.6073	0.0813	0.1212	0.5030	17.4821	0.9194	−4.7721	0.3803

**Table 4 foods-11-02643-t004:** Predicted and detected values (at t = 0 and 3 months) of responses for the plumcake realized under optimized formulation.

	Predicted Value	Detected Value	
	t = 0	t = 0	t = 3 Months
Hardness [N]	2.49	2.51 ± 0.02 ^a^	4.52 ± 0.02 ^b^
Porosity [%]	86.89	87.56 ± 0.03 ^a^	87.22 ± 0.02 ^b^
a_w_	0.81	0.81 ± 0.01 ^a^	0.78 ± 0.01 ^b^
Moistness	2.45	2.50 ± 0.02 ^a^	1.13 ± 0.01 ^b^

Different letters (^a, b^) reveal significant differences (*p* < 0.05) among the samples during the storage time.

## Data Availability

The data presented in this study are available on request from the corresponding author.

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
