# Peer review of "Olive Oil-Based Oleogel as Fat Replacer in a Sponge Cake: A Comparative Study and Optimization"

_foods, 2022, doi:10.3390/foods11172643_

Round 1

Reviewer 1 Report

This work investigated the replacement of butter with olive oil-based oleogel in plumcake formulation, proposing a mixture design approach for optimal product formulation. The topic is of great interest, however, the article needs to be completely re-structured in order to be considered for publication as in its current version it lacks a clear goal and is very hard to understand.

The article needs to be carefully revised as it contains numerous English mistakes and sentences that do not fully male sense. Also, there is no line number which makes it difficult to comment.  In general author need to make clear what is the main objective of the study. It is not at all clear for example what was the purpose of using whey protein in the recipe. Why would emulsifier and whey proteins avoid mold growth? I suggest abstract is re-written to improve clarity of what exactly why done. In addition, is there a specific reason why olive oil was chosen for oleogel formulation as compared to other vegetable oils olive oil has a characteristic taste and is generally more expensive? Moreover, author need to be consistent with the terminology used throughout the manuscript. For instance, the title says sponge cake and abstract traditional plumcake, authors should be consistent with the terminology that is used. Which one is it? No basic/common methodologies were used to evaluate neither the dough (apparent density and pH) neither the cake (moisture content, specific volume, and baking loss). For instance, dough density would be useful to see the role of fat addition. Is what author call moistness the moisture contect? If yes how was it measured and what unit?

Further comments:

Pg 2: Chemical leavening? Provide details for each of the ingredients used in the cake, brand for example. Provide details of which mono and diglycerides of fatty acids are present in E471

Pg 3: Provide reference for the “traditional plumcake” recipe? Is yogurt really used in a traditional recipe? To optimize the experiments or the variables?

Author Response

Answers to Reviewer 1

Manuscript ID: foods-1865889: Olive oil – based oleogel as fat replacer in a sponge cake: a comparative study and optimization

We gratefully acknowledge the Reviewer for his thorough and careful examination of the manuscript. We have considered his comments/suggestions and revised the paper accordingly. Additions in the manuscript are marked in red. We hope our answers can satisfy your right concerns and you can appreciate the whole study proposed in this manuscript

Reviewer

This work investigated the replacement of butter with olive oil-based oleogel in plumcake formulation, proposing a mixture design approach for optimal product formulation. The topic is of great interest, however, the article needs to be completely re-structured in order to be considered for publication as in its current version it lacks a clear goal and is very hard to understand.

We have edited the Abstract and the Introduction, with the aim to make the goal of the article clearer. Moreover, we have edited the text, trying to make the manuscript as accessible to the reader as possible.

The article needs to be carefully revised as it contains numerous English mistakes and sentences that do not fully male sense.

Dear Reviewer, the English language of the revised manuscript has been edited by a high expert English speaker.

Also, there is no line number which makes it difficult to comment.

We apologize for the lack of line numbering in the submitted manuscript, we have added the line numbering in the revised one.

In general author need to make clear what is the main objective of the study. It is not at all clear for example what was the purpose of using whey protein in the recipe. Why would emulsifier and whey proteins avoid mold growth?

In the revised manuscript we have added references explaining the role of emulsifiers and whey proteins as food additives/ingredients in bakery products  (lines 74-79). The following sentences has been added: “Mono and diglycerides of fatty acids (E471) may be used to overcome bread staling by creating emulsified complexes with amylose, ensuring a soft crumb structure during the storage, besides to act as preservatives [17]. Whey proteins, apart from being highly nutritious, are useful food ingredients in cakes [18,19]. In particular, whey proteins form gels capable of holding water, reducing free water available for mould growth [19], that is one of the major concerns about the shelf life of bakery products.

I suggest abstract is re-written to improve clarity of what exactly why done.

Abstract has been modified as required.

In addition, is there a specific reason why olive oil was chosen for oleogel formulation as compared to other vegetable oils olive oil has a characteristic taste and is generally more expensive?

The reason for the choice of olive oil for the oleogel production/formulation has been described in the revised manuscript as follow: “Thanks to its positive health effects due to fatty acids compositions (oleic, linoleic, linolenic acids) in addition to minor components (tocopherols and phytosterols), olive oil is one of the most promising vegetable oil for oleogel production [9].”.

Moreover, author need to be consistent with the terminology used throughout the manuscript. For instance, the title says sponge cake and abstract traditional plumcake, authors should be consistent with the terminology that is used. Which one is it?

We have modified the abstract and the manuscript highlighting that plumcake is a type of sponge cake leavened bakery product.

No basic/common methodologies were used to evaluate neither the dough (apparent density and pH) neither the cake (moisture content, specific volume, and baking loss). For instance, dough density would be useful to see the role of fat addition.

As the reviewer correctly pointed out, other methodologies (density, rheological parameters) are commonly applied to characterize bakery products' dough. This study aimed to optimise plum cake formulation, we have chosen, as reported by other papers on the matter , physical and sensory properties (hardness, porosity, water activity and moistness) as quality parameters of finished bakery products and thus as variables dependent of the D-optimal Mixture Design.

[Yilmaz, E.; Ogutcu, M. The texture, sensory properties and stability of cookies prepared with wax oleogel. Food Funct  2015, 6, 1194-1204;

 Khiabani, A.E.; Tabibiazar, M.; Roufegarinejad, L.; Hamishehkar, H.; Alizadeh, A. Preparation and characterization of carnauba wax/adipic acid oleo gel: A new reinforced oleo gel for application in cake and beef burger. Food Chem 2020, 333, 127446.]

 Is what author call moistness the moisture contect? If yes how was it measured and what unit?

We have added in the revised manuscript more information about moistness (lines 174-175) defined as the wet sensation of cake products during swallowing

The unit adopted rated on 5-point scales from “none” (1) to high” (5) as described in section “2.4.3 Sensory analysis” of the revised manuscript.

Further comments:

Pg 2: Chemical leavening? Provide details for each of the ingredients used in the cake, brand for example.

We have added in the revised manuscript (lines 100-104) the required information

Provide details of which mono and diglycerides of fatty acids are present in E471.

Dear reviewer, we don’t know the exact composition of the emulsifier E471, bought from the market. The information reported on the label by the company regarded only the definition “mono and diglycerides of fatty acids” in accordance with the Regulation (EC) n° 133/2008 on food additives. Commonly in the E471 emulsifier are present glyceryl monostearate and glyceryl distearate but we don’t know the ratio.

Pg 3: Provide reference for the “traditional plumcake” recipe? Is yogurt really used in a traditional recipe?

In the revised manuscript (lines 107-109) the following sentence has been added:

“The control plum cake recipe was obtained by preliminary experiments focused on the selection of the ratio among the ingredients listed in cake recipes books. The produced plumcakes were sensory evaluated by structure and taste.”

To optimize the experiments or the variables?

We have clarified the meaning of the sentence. In the revised manuscript (lines112-114) the following sentence has been reported: “A D-optimal mixture design was used to optimize the experiments which were carried out also to evaluate the effect of olive oil-based oleogel (x1), emulsifier E471 (x2) and whey proteins (x3) on the physicochemical and sensory properties of plumcakes.”.

Reviewer 2 Report

The paper describes an interesting set of experiments. The obtained data are correctly presented. The results are compared are related to previous similar work. Also, the results are interesting for their potential for practical application. Overall, I consider this to be an interesting paper that should be published. Nevertheless, I have a few concerns that should be addressed.

Specific points that the authors need to address:

2.1 Materials

Considering the basic ingredients of the plumcake dough, it would be appreciated if the authors indicate which type of flour and yogurt were used in the experiments.

3. Results and discussion

Experimental data for responses presented in Table 2 are given as a value ± (for example Hardness in P1 2.00±0.01). This indicates that eighter the experiments were replicated or which is more likely that the measurements were repeated. Please indicate this in the text how many repeat measurements were carried out.

Figure 2.

I suggest to authors to zoom in the part of the 2D-contour plots (lower left corner) which actually show the effect of oleogel, emulsifier E471 and whey proteins on measured responses. Currently, this part of the figure is quite small and not visible enough.  

Figures 3. and 4. and Table 4.

Authors stated that different letters (a,b,…) reveal significant differences (p<0.05) among the samples. It is not stated in the text which statistical method was used to test the significance of the difference between different formulations (P1-P10) and control (C). Also, based upon how many repeat measurements this was done?

Figure 5

The sensory profile plumcake samples shown in figure 5. includes formulations P11 and P12 which haven’t been mentioned previously anywhere in the paper. They just suddenly appeared for Figure 5 without any explanation in the text. This must be corrected either by giving a figure without P11 and P12 or by explaining formulations P11 and P12 and what authors wanted to achieve by including them in the Spider plots of sensory attributes.

3.5 Mixture optimization to produce plumcake with desired characteristics

One of the aims of this research or even the main goal of this research was to identify the optimum formulation as it was clearly stated by the authors themselves in Title, Abstract, Introduction, Results and Discussion. Therefore, I don’t find satisfactory just to say “The optimum percentages of ingredients, as generated by the JMP software through the desirability function, were 76.98% olive oil-based oleogel, 7.28%, emulsifier E471 and 15.73%, whey proteins.” and “obtaining a product with textural and sensory properties close to those existing in the plumcakes market leader.” What were the optimization conditions, that is, what were the target values ​​for the outputs? Were they set to a maximum, minimum, or within an interval of values? Are any of the outputs were given priority, etc. The optimization conditions must be clearly stated.

Table 4. Predicted and detected values (at t=0 and 3 months) of responses for plumcake realized under optimized formulation.

The numbering of this table should be changed to Table 5 considering that Table 4 already exists - Table 4. aw evaluation for control and butter replaced samples. Also, this change should be followed in the text when referring to the data from the corresponding table: „Finally with the aim to verify the efficacy of formulation able to preserve the investigated quality parameters in table 4 were reported hardness, porosity, aw and moistness before and after 3 months of storage.“

Please identify which statistical method was used to compare the results of detected values at t=0 and t=3 months.

Author Response

Answers to Reviewer 2

Manuscript ID: foods-1865889: Olive oil – based oleogel as fat replacer in a sponge cake: a comparative study and optimization

We gratefully acknowledge the Reviewer for his thorough and careful examination of the manuscript. We have considered all his comments/suggestions and revised the paper accordingly. Additions in the manuscript are marked in red 

The paper describes an interesting set of experiments. The obtained data are correctly presented. The results are compared are related to previous similar work. Also, the results are interesting for their potential for practical application. Overall, I consider this to be an interesting paper that should be published. Nevertheless, I have a few concerns that should be addressed.

Specific points that the authors need to address:

2.1 Materials

Considering the basic ingredients of the plumcake dough, it would be appreciated if the authors indicate which type of flour and yogurt were used in the experiments.

The required information has been added to the revised manuscript (lines 100-102)

  1. Results and discussion

Experimental data for responses presented in Table 2 are given as a value ± (for example Hardness in P1 2.00±0.01). This indicates that eighter the experiments were replicated or which is more likely that the measurements were repeated. Please indicate this in the text how many repeat measurements were carried out.

Thank you for the suggestion. The required information has been added to the revised manuscript (lines 166-167)

Figure 2.

I suggest to authors to zoom in the part of the 2D-contour plots (lower left corner) which actually show the effect of oleogel, emulsifier E471 and whey proteins on measured responses. Currently, this part of the figure is quite small and not visible enough. 

Thank you for the suggestion. In figure 2, 2D-contour plots have been zoomed as required.

Figures 3. and 4. and Table 4.

Authors stated that different letters (a,b,…) reveal significant differences (p<0.05) among the samples. It is not stated in the text which statistical method was used to test the significance of the difference between different formulations (P1-P10) and control (C). Also, based upon how many repeat measurements this was done?

The suggested information has been added to the revised manuscript (lines 200-203).

Figure 5

The sensory profile plumcake samples shown in figure 5. includes formulations P11 and P12 which haven’t been mentioned previously anywhere in the paper. They just suddenly appeared for Figure 5 without any explanation in the text. This must be corrected either by giving a figure without P11 and P12 or by explaining formulations P11 and P12 and what authors wanted to achieve by including them in the Spider plots of sensory attributes.

Sorry for the mistake. We have added the right figure to the revised manuscript (line 336).

3.5 Mixture optimization to produce plumcake with desired characteristics

One of the aims of this research or even the main goal of this research was to identify the optimum formulation as it was clearly stated by the authors themselves in Title, Abstract, Introduction, Results and Discussion. Therefore, I don’t find satisfactory just to say “The optimum percentages of ingredients, as generated by the JMP software through the desirability function, were 76.98% olive oil-based oleogel, 7.28%, emulsifier E471 and 15.73%, whey proteins.” and “obtaining a product with textural and sensory properties close to those existing in the plumcakes market leader.” What were the optimization conditions, that is, what were the target values ​​for the outputs? Were they set to a maximum, minimum, or within an interval of values? Are any of the outputs were given priority, etc. The optimization conditions must be clearly stated.

Thank you for thesuggestion. The following sentence has been added to the revised manuscript (lines 194-199):“ In particular, the optimum formulation has been obtained by setting the goal of the desirability functions for the hardness and water activity to a minimum value and for porosity and moistness to a maximum value in established value ranges, through the measurement of physical and sensory parameters of plumcakes market leader. The other sensory properties (sweetness, flavour, off flavour, and overall acceptability were used for the global evaluation of the final product.

Table 4. Predicted and detected values (at t=0 and 3 months) of responses for plumcake realized under optimized formulation.

The numbering of this table should be changed to Table 5 considering that Table 4 already exists - Table 4. aw evaluation for control and butter replaced samples. Also, this change should be followed in the text when referring to the data from the corresponding table: „Finally with the aim to verify the efficacy of formulation able to preserve the investigated quality parameters in table 4 were reported hardness, porosity, aw and moistness before and after 3 months of storage.“

Please identify which statistical method was used to compare the results of detected values at t=0 and t=3 months.

Sorry for the mistake. We have updated the numbering of the tables and the table references in the manuscript. As regards the comparison of the results between plumcakes at t=0 and t=3 months the employed statistical method has been described in the section "2.5 Statistical analysis" of the revised manuscript (lines 200-203).

Reviewer 3 Report

The article is very well put together and is easy to read and understand. I only recommend correcting some details or finger errors indicated in the attached document.

Author Response

Answers to Reviewer 3

Manuscript ID: foods-1865889: Olive oil – based oleogel as fat replacer in a sponge cake: a comparative study and optimization

We gratefully acknowledge the Reviewer for his thorough and careful examination of the manuscript. We have considered all his comments/suggestions and revised the paper accordingly.

The article is very well put together and is easy to read and understand. I only recommend correcting some details or finger errors indicated in the attached document.

Delete 00

We have changed  “flour  00” in  “wheat flour type 00”( line 100)

I recommend to change colors or line patterns to enhance the properly identification of the attributes for each of the formulations

We have modified the figure 5 as you suggested

Round 2

Reviewer 1 Report

""The article is considerably improved, and the aim is now clear.  

However, there are still some details authors should consider. For instance, sometimes author write plumcake and sometimes plum cake, authors should stick to the use of only one terminology/format. Maybe the title could also be changed to plum cake instead of sponge cake?

Line 15: One cannot affirm that every oleogel does not affect the quality of the final product as this for sure depend on the ingredients used for oleogel products and the interaction of the oleogel with food matrix

Line 54: Many other oil types also have good composition and are also promising. Authors should only mention it is a good alternative and not necessarily “the most promising”

Line 172: Punctuation

Line 260: who observed instead of “which”

Table 4 is useless as it contains the exact same data as in table 2 (except it contains also data for control). Therefore, I suggest to remove table 4, reference table 2 for this data and include result for the control in table 4 or within the text.

Author Response

Answers to Reviewer 1

Manuscript ID: foods-1865889: Olive oil – based oleogel as fat replacer in a sponge cake: a comparative study and optimization

We gratefully acknowledge the Reviewer for his thorough and careful examination of the manuscript. We have considered his comments/suggestions and revised the paper accordingly. Additions and changes in the manuscript are marked in red.

Reviewer

 The article is considerably improved, and the aim is now clear.  

Dear reviewer thank you, we appreciate your comment.

However, there are still some details authors should consider. For instance, sometimes author write plumcake and sometimes plum cake, authors should stick to the use of only one terminology/format.

We apologize for the mistake “plum cake” that has been changed in “plumcake” in line 106 of the revised manuscript.

Maybe the title could also be changed to plum cake instead of sponge cake?

Dear reviewer, we prefer to maintain the term “sponge cake” in the title, because the oleogels and the D-optimal mixture design employed in this study can be adapted to all types of sponge cake bakery products and not only to plumcakes.

Line 15: One cannot affirm that every oleogel does not affect the quality of the final product as this for sure depend on the ingredients used for oleogel products and the interaction of the oleogel with food matrix

We have modified the text in the revised manuscript line 15 as follows: “Oleogels (defined as structured solid-like materials with a high amount of oil entrapped within a three-dimensional network of gelator molecules) represent a healthy alternative to fats rich in saturated and trans fatty acids”

Line 54: Many other oil types also have good composition and are also promising. Authors should only mention it is a good alternative and not necessarily “the most promising”.

Dear Reviewer, we know that other types of vegetable oils have a good composition for human health. In fact, in the manuscript, we assert that olive oil is “one of the most promising vegetable oil for oleogel production”.

Line 172: Punctuation

We apologize for the mistake. We have deleted the punctuation in the line 171.

Line 260: who observed instead of “which”

We have changed “which” with “who” as you rightly suggested.

Table 4 is useless as it contains the exact same data as in table 2 (except it contains also data for control). Therefore, I suggest to remove table 4, reference table 2 for this data and include result for the control in table 4 or within the text.

As suggested, we have removed table 4. The text in the revised manuscript (lines 324-327) has been modified as follows: ”The comparison of aw values among the control (0.83±0.01) and all butter replaced formulations (table 2) showed no significant (p<0.05) differences except for the sample containing 72% olive oil-based oleogel, 3% emulsifier and 25% whey proteins that showed the lowest aw value of 0.79.
